# Sunlight penetration dominates the thermal regime and energetics of a shallow ice-covered lake in arid climate

Wenfeng Huang[1,2]\*, Wen Zhao[1], Cheng Zhang[1], Matti Leppäranta[3], Zhijun Li[4]\*, Rui Li[1], Zhanjun Lin[2]

Key Laboratory of Subsurface Hydrology and Ecological Effects in Arid Region (the Ministry of Education), Chang'an University, Xi'an, China
State Key Laboratory of Frozen Soil Engineering, Northwest Institute of Eco-Environment and Resources, Chinese Academy of Science, Lanzhou, China
Institute of Atmospheric and Earth Sciences, University of Helsinki, Helsinki, Finland.
State Key Laboratory of Coastal and Offshore Engineering, Dalian University of Technology, Dalian, China

*Correspondence to: Wenfeng Huang* (huangwenfeng@chd.edu.cn) *and Zhijun Li* (lizhijun@dlut.edu.cn)

**Abstract.** The Mongolian Plateau is characterized by cold and arid winters with very little precipitation (snowfall), strong solar insolation, and dry air, but little is known about the thermal regimes of the ice and ice-covered lakes and their response to the distinct weather and climate in this region. In a typical large, shallow lake, ice and snow processes (cover) and under-ice thermodynamics were monitored for four winters in 2015–2019. Heat transfer at the ice-water interface and lake heat budget were investigated. The results revealed that persistent bare ice of 35–50 cm thickness transmits 20–35% of the incident solar radiation into the water below. This is a dominant source for under-ice energy flows and causes/maintains high water temperature (up to 6–8°C) and high heat flux from water to ice (averages of 20–45 W m$^{-2}$) in mid-winter, as well as higher heat conduction in the ice interior during freezing. The heat balance shows that the transmitted radiation and the heat flux from water to ice are the dominant and highly correlated heat flows in the lake. Both bulk water temperature and temperature structure are sensitive to solar transmittance and occasional snow events. Under-ice convective mixing does not necessarily occur because of stratification of salinity in the water body. In particular, salt exclusion during freezing changes both the bulk salinity and the salinity profile, which plays a major role in the stability and mixing of the water column in this shallow lake.

## 1 Introduction

Lakes are important water resources and provide vital habitats for aquatic ecosystems. More than 55% of world lakes are located between 40 and 80°N in the northern hemisphere (Verpoorter et al., 2014) and have potential to freeze seasonally (Kirillin et al., 2012), especially in Arctic, boreal, and temperate climate, and high mountain regions. Due to distinct properties of ice compared to water, seasonal formation and decay of ice cover have tremendous impacts on the lake water quality (Yang et al., 2016), physical and chemical conditions (Yang et al., 2021; Cavaliere and Baulch, 2018; Huang et al., 2019a), aquatic ecosystem (Griffiths et al., 2017; Song et al., 2019), and land-atmosphere mass and heat interaction (Wang et al., 2015; Franz et al., 2018). Therefore, common concerns have been widely spread on mapping lake ice physics and its underlying physical mechanisms in the evolution of ice seasons.

Field and modeling investigations on lake ice processes have a long history in northern temperate and boreal regions, such as Fennoscandia, central Europe, northern Canada, and the Great Lakes. Shortening of ice cover period has been documented currently in lakes in these northern regions (Bernhardt et al., 2012; Lei et al., 2012; Karetnikov et al., 2017; Ptak et al., 2020). However, the lake ice regime has remained less studied due to lack of long-term observational records in arid climate regions, such as central Asia and high mountains, which are subject to a quite different landscape, regional climate, and hydrological cycles compared with the northern temperate, boreal, and Arctic environment.

Lake thermal stratification is of great importance to hydrodynamics and transport of nutrients, oxygen, and primary production, which influence limnological habitats and ecosystems. In freezing freshwater lakes, stable inverse thermal stratification usually forms and persists under the ice cover with the temperature smaller than the maximum density temperature (3.98°C). After the onset of melting, strong solar irradiance can penetrate the ice cover into the water and drive turbulent convection (Bouffard et al., 2019; Volkov et al., 2019) until the bulk temperature reaches or surpasses the maximum density temperature or until breakup (Yang et al., 2020). However, in some shallow mid-latitude brackish lakes, this is not the story. During melting period, a warm middle layer may form due to salinity stratification and separate the overlying inverse thermal stratification and the underlying positive thermal stratification. The temperature of this warm layer can go up to around 10°C before the breakup (e.g. Huang et al., 2019b; Kirillin et al., 2021). This underlines the uniqueness of seasonally ice-covered lakes in mid-latitude arid regions and the importance of their different climate. It still remains unclear how this stratification forms and evolves and how it interacts with the snow/ice cover.

After freeze-up, the ice cover shelters the lake from atmospheric forcing and deposits. The lake boundary constitutes of only the ice cover on the top and sediment at the bottom. The heat budget is governed by radiative and sensible fluxes across the ice-water-sediment interfaces (Leppäranta, et al., 2019). But these fluxes, including solar radiation transfer, ice-water heat exchange, and sediment heat release, have not been well quantified in mid-latitude arid region lakes. Especially, the ice-water heat flux, a key factor affecting the mass and energy balance of both ice and water, has been demonstrated to be remarkably higher in Central Asia than in Arctic and boreal zones (Malm et al., 1997; Jakkila et al., 2009; Huang et al., 2019a,b; Lu et al., 2020). But the regime underpinning its high values is still unknown.

To fill the knowledge gaps in winter thermodynamics of lakes in cold and arid Asia and their background energy flows, we performed a four-winter observation program of snow/ice processes, solar radiation transfer, and temperature profiles of air-ice-water-sediment column in a typical large shallow lake that is seasonally ice-covered for 4–5 months, located in the southern border of the Mongolia Plateau. Below, observations and models are combined 1) to reveal the seasonal and diurnal dynamics of the temperature stratification under ice in the mid-latitude arid climate, and 2) to quantify and balance the involved heat fluxes that determine the thermal state of the lake.

**2 Data and methods**

**2.1 Study site**

The Hetao Basin (ca. 6,000 km$^2$, mean altitude > 1,000 m), one of the oldest and largest irrigation areas in China, is located in the central southern Mongolian Plateau controlled by temperate continental climate. In the Hetao Basin, the annual sunshine duration is 3,000–3,200 h, the annual air temperature is 5.6–7.4°C with the lowest and highest monthly temperature of –14– –11°C (Jan) and 22–24°C (Jul), the frost-free period is 130–150 d, and the annual precipitation is 150–400 mm concentrated in the warm season. Most parts of the basin have been desertified or semi-desertified in recent decades.

Lake Ulansuhai (40°36′–41°03′N, 108°41′–108°57′E, altitude 1,019 m) is a typical large, shallow lake
in desert/semi-desert region with a total area of about 306 km² (Fig. 1). It is a very important part of the
irrigation and drainage system of the Hetao Basin, and its major water source comes from the farmland
irrigation drainage and domestic sewage. The maximum and mean depths are 2.5–3.0 m and 1.0–1.5 m,
respectively. The annual air temperature, hours of sunshine, precipitation, evaporation, wind speed,
frost-free period are 7.3°C, 3,185 h, 224 mm, 1,502 mm, 3.5 m s$^{-1}$, and 152 d, respectively (Sun et al.,
2011). The solar noon-time and altitude in winter are 12:45±15 min and 41±10°, respectively. The ice
cover is usually free of snow or only sparsely snow-covered due to occasional snowfall events and
strong winds.
The lake surface elevation is regulated through pumping water from the Yellow River via the main
inflow canal at the western shore. The total annual water supply is approximately $4 \times 10^8$ m³,
equivalent to the lake volume. But in winter (Nov–Mar), very little surface inflow/outflow exists
except possible minor wastewater inflow (Sun et al., 2013), and the subsurface inflow is also negligible
(Zhu et al., 2014). For more detailed information, please consult Lu et al. (2020) and references therein.
According to our sampling tests in winter 2017, the lake water is brackish or weakly saline with
salinity of 1.0–1.5‰ before the ice-on and gradually increasing to 2.5–3.0‰ due to exclusion of salts
when the ice grows to its annual maximum.
Due to its unique climate and eutrophication, the lake ecology under the ice cover is very active with
high rates of primary production and respiration. This is believed to be highly related to the under-ice
solar irradiance and temperature and the key role of ice and snow processes (Song et al., 2019; Huang
et al., 2021). Our previous observations revealed the mass and heat balance of the lake ice cover and
the impacts of warm water under the ice cover (Lu et al., 2020), but further investigations were
performed and combined here to look into the thermal stratification regimes.

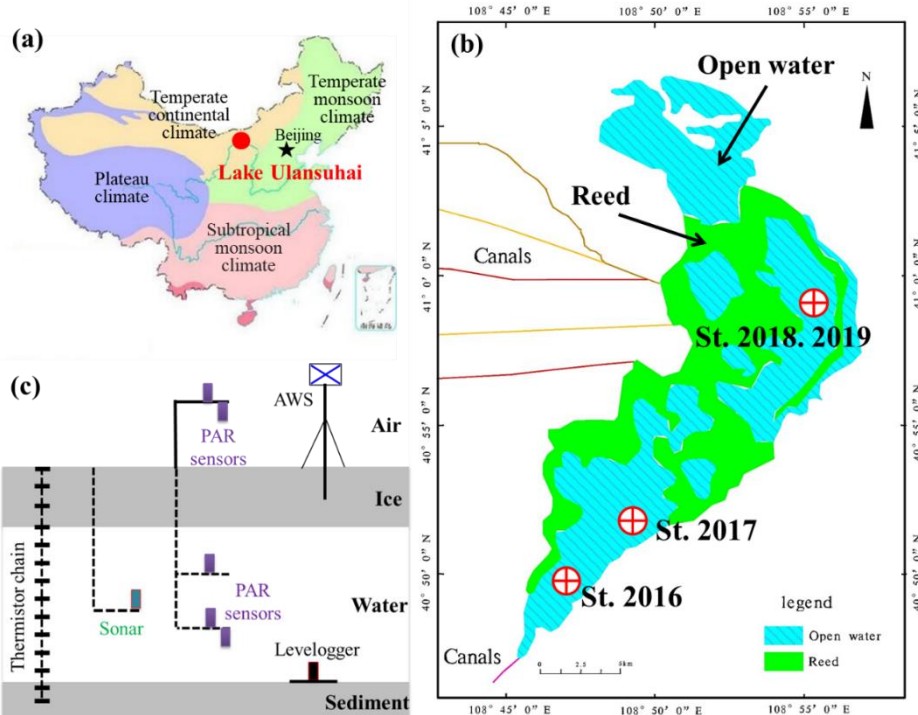


**106  Figure 1. Locations of Lake Ulansuhai (a) and study sites (b) and the field instrumentation (c). In**

**107  each winter, a thermistor chain was refrozen into ice cover to measure temperature profile of the**

**108  air-ice-water-sediment column with an established automatic weather station (AWS); more than 3**

**radiation sensors over photosynthesis active radiation (PAR) band were deployed to observe the incident, reflected and transmitted solar radiation.**

**2.2 Data acquisition**

During winters of 2015-2019, field campaigns were conducted in open, reed-free areas of Lake Ulansuhai (Figs. 1c and 2). In each winter, an automatic weather station (AWS) was established on the ice cover, to record wind speed and direction, air temperature and humidity, incident and reflected global radiation (300–3000 nm), and the skin temperature of the ice/snow surface. An under-ice uplooking sonar (WUUL-1/2, Wuhan University, China) was used to measure the ice thickness evolution with the accuracy of 2 mm. Snow thickness was measured manually using a snow stake every 1–2 days. The temperature profile of the air-ice-water-sediment column was observed using a thermistor chain (PTWD, Jinzhou Sunshine Technology Co. Ltd, China) at 5–10 cm spacing with the accuracy of 0.05°C. TriOS spectral radiometers with the accuracy of 0.04–0.06 mW m$^{-2}$ nm$^{-1}$ (RAMSES-ACC-VIS, TriOS, German) were used to measure the incident and reflected photosynthetically active radiation (PAR) over the ice/snow surface and under the ice cover. The water level was monitored using a temperature-pressure logger with an accuracy of 0.05% (LTC Levelogger, Solinst, Canada) placed 20 cm above the sediment surface. All the above variables were recorded every 10 min. Information of the acquired datasets is summarized in Table 1 (see also Huang et al. 2021).

In the winter of 2017, the under-ice water electric conductivity (EC) was measured using 3 online conductivity loggers (HOBO U24, Onset, USA, accuracy of 3%) at depths of 60 cm, 100 cm, and 150 cm from Jan 21 to Mar 11 (Table 1). Concurrently, ice and water samples with 5 cm spacing were collected 8 times this winter to measure their EC and salinity using a portable YSI salinometer with accuracy of 1%.

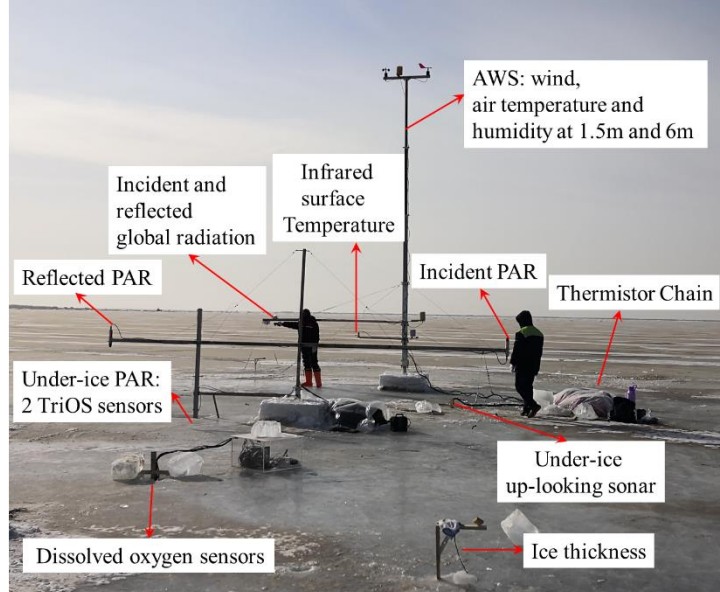

**Figure 2. Field setup of apparatus deployed and corresponding monitored variables in winter of 2019**

**Table 1.** Data series acquired during the four-winter observation program

| Winter | 2016 | 2017 | 2018 | 2019 |
|---|---|---|---|---|
| Available | Jan 11–Mar 9 | Jan 21–Mar 11 | Jan 9–Feb 25 | Jan 20–Feb 27 |

| | | | | |
|---|---|---|---|---|
| duration | | | | |
| Water depth | 220 cm | 170 cm | 143 cm | 140 cm |
| Ice/snow thickness | √ | √ | √ | √ |
| Air-ice-water-sediment temperature | 5 cm spacing in ice, 10–15 cm spacing in water and sediment | 5 cm spacing in ice, 5–10 cm spacing in water and sediment | 5 cm spacing in ice, 5–10 cm spacing in water and sediment | 5 cm spacing in ice, 10 cm spacing in water and sediment |
| Under-ice irradiation | 175 cm | 80 cm, 130 cm | 65 cm, 90 cm, 120 cm | 80 cm, 120 cm |
| Under-ice upwelling radiation | | 105 cm | 100 cm | |
| Water level | | √ | | |
| Electric Conductivity | | 60 cm, 100 cm, 150 cm | | |

Note: the observed depths for under-ice irradiation and upwelling radiation and electric conductivity
denote the distances below the ice surface.
**2.3 Heat flux calculation and balance**
In freshwater lakes, the water temperature is colder than 3.98°C with a weak inverse thermal stratification
during freezing (Winter I stage), and typically a convective mixing layer forms between the top cold
interfacial layer and the warm quiescent layer during melting (Winter II stage) (Kirillin et al., 2012). The
stratification structure in Lake Ulansuhai was checked using temperature gradient following Kirillin et
al. (2018).
After freeze-up, as illustrated in Fig. 3, the thermal regime of the water column is governed by the solar
irradiance penetrating through the ice cover ($R_w$), solar radiation absorbed by the lake sediment (if any)
($R_{sed}$), heat fluxes through ice-water ($F_w$) and water-sediment ($F_{sed}$) interfaces, and horizontal heat gain
by advection and diffusion ($F_h$) from the neighboring water body. If the zero-reference level for heat is
defined as the heat content of liquid fresh water at its freezing point temperature, the heat content of
water is $\rho_w c_w T_w h_w$, and the heat budget of a water column is
$$R_w - R_{sed} + F_{sed} + F_h - F_w = \rho_w c_w h_w \frac{dT_w}{dt} + \rho_w c_w T_w \frac{dh_w}{dt}, \quad (1)$$
where $\rho_w$, $c_w$, and $T_w$ are the density, specific heat, and bulk temperature of water, respectively. Other
variables are defined in Fig. 3. The lateral heat transport $F_h$ is negligible in this shallow lake with a flat
bottom (Rizk et al., 2014; Kirillin et al., 2015). The two terms on the right-hand side are the heat content
changes induced by changes in the water temperature and depth, respectively. The water level logger
indicated that the lake lost water through seepage to soil quite slowly (about 0.6 mm/d) during ice seasons,
and the heat loss due to the bottom water seepage was estimated to be smaller than 0.8 W m$^{-2}$ and thus
was ignored as a minor term compared to the other heat fluxes.
*Under-ice solar irradiance.* The light extinction coefficient of the under-ice water was measured as 2.1
m$^{-1}$ under a clear sky on Jan 7, 2018. Using the observed irradiance by under-ice spectral sensors, the
solar irradiance at the ice-water interface ($R_w$) was derived from a one-band exponential decay law of
light transfer in water column, following
$$R_w = R_d \exp(\kappa_w(z_d - h_i)), \quad (2)$$
where $R_d$ is the observed downward irradiance at depth $z_d$, $h_i$ is the ice thickness, and $\kappa_w$ is the light
extinction coefficient of water.
*Sediment heat flux.* The heat exchange flux through the water-sediment interface ($F_{sed}$) was calculated
with the gradient method,
$$F_{sed} = -\kappa_{sed} \left. \frac{\partial T_{sed}}{\partial z} \right|_{bottom} \approx -\kappa_{sed} \frac{\Delta T_{sed}}{\Delta z}, \quad (3)$$
where $\kappa_{sed}$ is the thermal conductivity of sediment and $T_{sed}$ is the observed sediment temperature. In the
winter of 2018, four thermistors were buried in the sediment (1 cm, 9cm, 17 cm, and 30 cm below the
sediment surface) to measure the sediment temperature profiles. Assuming the heat transfer in the top
sediment is governed by the typical one-dimensional vertical heat conduction equation, an optimal
control model was deployed to retrieve the effective thermal diffusivity of the sediment was estimated
based on the observed sediment temperature profiles. For details on the optimal control model, please
refer to Shi et al. (2014). The thermal conductivity can be determined (0.2–0.7 W m$^{-1}$°C$^{-1}$) with measured
density and specific heat capacity of sediment. $\kappa_{sed}$= 0.5 W m$^{-1}$°C$^{-1}$ was used in Eq. 3.
*Water-to-ice heat flux.* The water-to-ice heat flux can be derived from the heat balance at the ice-water
interface,
$$F_w = Q_c - Q_l = -\kappa_i \left. \frac{\partial T_i}{\partial z} \right|_{z=h_i} - \rho_i L_f \frac{\partial h_i}{\partial t}, \quad (4)$$
where $Q_c$ and $Q_l$ are the conductive heat flux to ice and the latent heat due to freezing/melting,
respectively, and $\rho_i$ and $L_f$ are the density and latent heat of fusion of ice, respectively. The first term
denotes the heat conduction into the ice interior, which can be derived using the temperature gradient in
the bottom ice layer. The second term on the right-hand side denotes the heat release/absorption due to
freezing/melting, which can be derived from ice thickness observations.
The heat content due to temperature change (i.e., the first term on the right-hand side of Eq. 1) was
calculated using the observed water temperature profiles.
Direct use of semi-hourly observed datasets brought high-frequency fluctuations in estimated heat flux,
and then daily means were used for further analysis of seasonal dynamics.

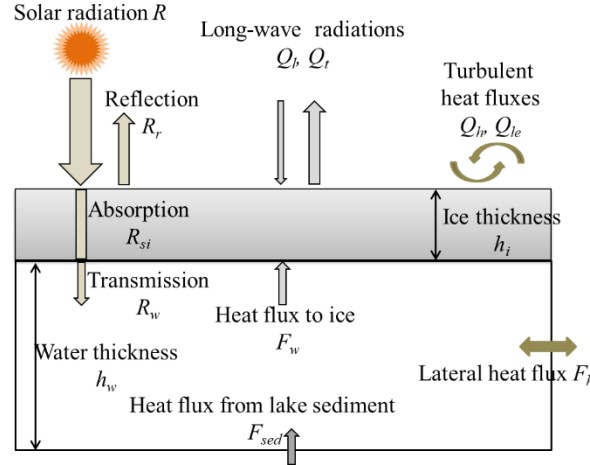

**Figure 3. Heat budget components of the water column under the ice (modified from Huang et al,**
**2019b).**

### 2.4 Potential errors in heat flux estimation

Potential errors in the above heat flux estimation usually come from the measurement accuracy of the
deployed apparatuses. The maximum error for each flux was determined based on the related apparatus
accuracy (Table 2). The thermistor accuracy is expected to lead to errors less than 1.7 W m$^{-2}$ on $F_s$ and
$F_{Tw}$, and the ice density caused errors less than 1.3 W m$^{-2}$ on $F_w$. Other heat fluxes suffer to only negligible
uncertainties ($< 0.3$ W m$^{-2}$) induced by individual sources. Errors from several sources may accumulate
especially in $F_w$, but the accumulated errors in $F_w$ should be less than 8%.

**Table 2.** Uncertainties in calculation of heat fluxes

| Error source | Errors in heat flux (W m$^{-2}$)* | | | |
|---|---|---|---|---|
| | $R_w$ | $F_{sed}$ | $F_w$ | $F_{Tw}=\rho_w c_w h_w \dfrac{dT_w}{dt}$ |
| Radiation precision | 0.08 | – | – | – |
| Thermistor precision | – | 0.25 | 1.1 | 1.7 |
| Ice thickness | 0.1 | – | – | 0.01 |
| Ice growth rate | – | – | 0.3 | – |
| Ice density | – | – | 1.25 | – |
| Water density | – | – | – | 0.2 |

* Dashes (–) indicate inapplicable.

## 3 Results

### 3.1 Lake ice and temperature

Our observations were conducted during mid-winter covering the turning point from ice growth to
melting. The air temperature was consistently lower than 0°C, but its daily amplitude was very high (10–
16°C) and the peak at noon/afternoon could be close to 0°C (Fig. 4). Wind speed was generally lower
than 4 m s$^{-1}$ except occasional gusts that led to snow or dust drifting. The relative humidity of air 2-m
above the ice surface also showed an evident diurnal cycle between 40% and 80%.
The peak incident solar radiation was each day roughly 500–800 W m$^{-2}$, and its daily average was 80–
200 W m$^{-2}$ showing an increasing trend from the beginning to the end of our observation period. But the
daily average was always smaller than 100 W m$^{-2}$ due to prevailing cloudy or overcast skies in winter
2019. Occasional snowfalls usually brought thin snow layers ($< 6$ cm) that continuously ablated due to
wind blowing and melting and sublimation. A new snow cover could increase the surface albedo up to
over 0.80 but this increment gradually disappeared within one week following the snowfall.
The maximum annual ice thickness varied between 35 cm and 60 cm, accounting for 30%–60% of the
mean lake depth. The bulk water temperature under ice cover was 3–7°C and showed diurnal cycles and
synoptic decreases following snowfall events, evidencing the decrease of transmitted solar radiation. The
sediment surface layer was always warmer than the water column during the observation period, showing
that the sediment works as a heat source to the overlying water.

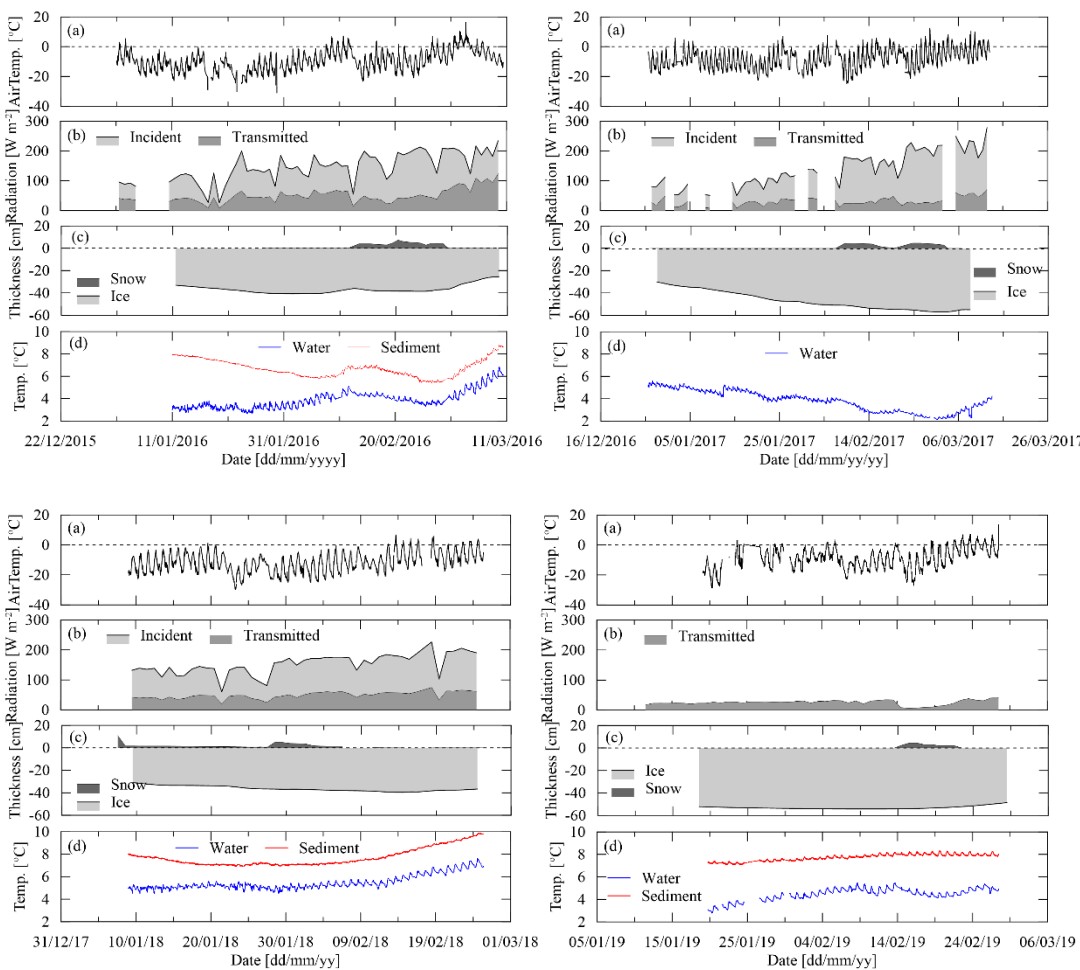

**Figure 4. Observational air temperature *Ta* (a), daily means of incident and transmitted solar radiation (b), snow and ice thickness (c), temperature of water column and surface sediment. (top left: winter 2016; top right: winter 2017; bottom left: winter 2018; bottom right: winter 2019)**

## 3.2 Thermal stratification and mixing in midwinter

In mid-winter, the lake sediment was still very warm with surface temperature > 6°C, usually causing temperatures higher than 4°C in the lower part of water column (Fig. 5). It is hypothesized that this stratification was supported by salinity stratification. There is no detailed concurrent salinity profile data available, but the bulk salinity is of the order of 1‰, enriched in ice season. As was observed in the winter of 2017 (Fig. 6), stable salinity stratification existed during the freezing period, and as salt was continuously excluded to water through the ice-water interface during water freezing, the bulk salinity increased and the salinity structure approached gradually to neutral stratification. But when the ice melting began on Mar 7, the bulk salinity decreased and stable salinity stratification formed again due to fresher meltwater intrusion. The sensitivity of density to temperature is very low in the neighborhood of 4°C so quite small salinity changes can compensate for the observed temperature structure for neutral density stratification. Although our observations didn't cover the whole ice season, evident seasonal and annual variations were observed.

A common thin layer (10–30 cm) of strong inverse stratification (i.e., interface layer) prevailed just beneath the ice due to the large difference in temperature of the ice base at the freezing point and the

bulk water column, e.g. in winters 2016, 2018 and 2019. But in winter 2017, this thin top layer did not show up and a persistent thick inverse structure developed through the water column (Fig. 5b). Underneath the top cold interface layer, the temperature increased slowly downward to the warm sediment (weak inverse structure) in winter 2019 and prior to 3 Mar in winter 2016 (Fig. 5a and 5d). After 3 Mar in winter 2016 and 10 Feb in winter 2018, a thermally homogeneous convective layer quickly developed after the bulk water temperature rose above approximately 7°C (Fig. 5a and 5c). Strikingly, before the formation of convective mixing in winter 2018, a "warm" zone of 30 cm (local maximum temperature) with temperature decreasing both downwardly and upwardly persisted at ~30 cm beneath the ice base. This abnormal layer is sometimes called a local temperature minimum (Mironov et al., 2002) or a "temperature dichotomy" (i.e., a dicothermal layer used in oceanography) (e.g., Kirillin et al., 2011, 2021). Water temperature contours (not shown) revealed that both the bulk temperature and thickness of the dicothermal layer show significant diurnal cycles: its temperature and thickness take up and increase following the solar insolation cycle and decrease or even disappear during the night. The development and extension of this layer also increase the thermal gradient of the overlying interface layer.

Occasional snowfall events usually led to quick bulk cooling along the entire water temperature profile due to the high reflection of new snow despite their small thickness. The sensitive response of water temperature to snow events (actually changes in penetrated radiation) implies large heat flux from water to ice and the dominance of solar radiation in this lake.

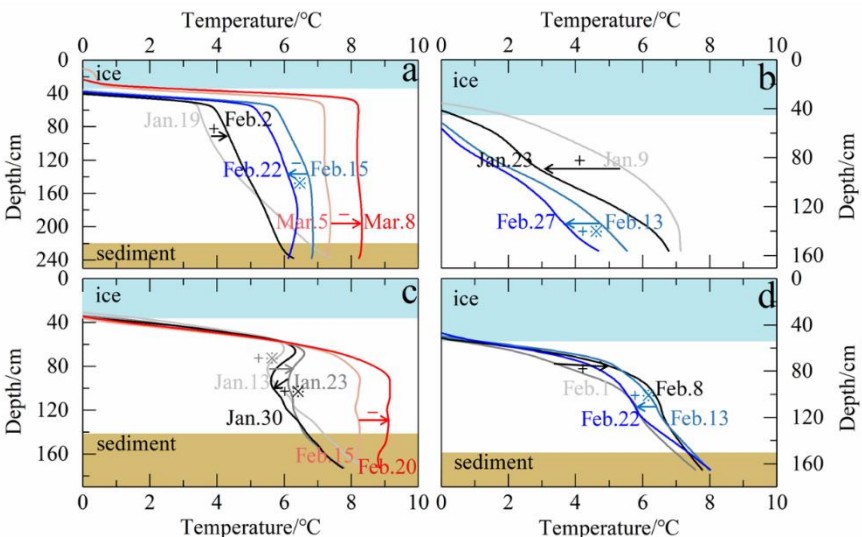

**Figure 5. Daily profile evolution of water column during ice season of winters (a) 2016, (b) 2017, (c) 2018, and (d) 2019. Light blue and brown zones denote ice cover and bottom sediment, respectively. Asterisks (※) denote snowfalls and snow-covered periods. Plus (+) and minus (–) denote the growth and melt stage of the ice cover.**

Unconventionally, under-ice convection did not take place in all winters (only two of our four observational winters) and seems to develop just when the bulk water temperature goes up to 7°C. This temperature threshold is higher than the temperature of maximum density of freshwater (3.98°C) and saline water (<3.98°C). These annually variable convections are believed to form conditionally and lake-specifically with proper water-sediment temperature and salinity profile. When the water temperature is large enough, its density effect overcomes salinity stratification and convection is thus triggered. Taking winter of 2017 as an example, water sampling indicated that, in this very shallow lake, the salinity

increased and its profile structure changed simultaneously as the ice grew (Fig. 6). At the ice-on, the salinity showed a stable profile (increasing downwardly) and its impact on water density outweighed the impact of concurrent temperature gradient (i.e., on Jan 5). With the following ice growth, the bulk salinity increased but the salinity gradient decreased, and the temperature gradient decreased. Consequently, the weakened salinity gradient could persistently outweigh or offset the impact of temperature profile on water density through the growing period (before Mar 4). Otherwise, if the weakening gradient of salinity no longer offsets the temperature effect, the convective mixing takes place across the density instability layer. This is very likely why under-ice mixing occurred in the winters of 2016 and 2018. When the ice started melting, the salinity gradient turned larger due to fresh meltwater released from the top, the water column or the top layer became more stable (on Mar 11).

We can conclude that how the water temperature and salinity profiles change synchronously during late freezing and initial melting determines whether the under-ice convection takes place. Especially, if the sediment temperature is high and the transmitted radiation is large during freezing, the sediment and bottom water temperature can be warm and increase rapidly, increasing the probability for full-depth convection such as in the winters of 2016 and 2018.

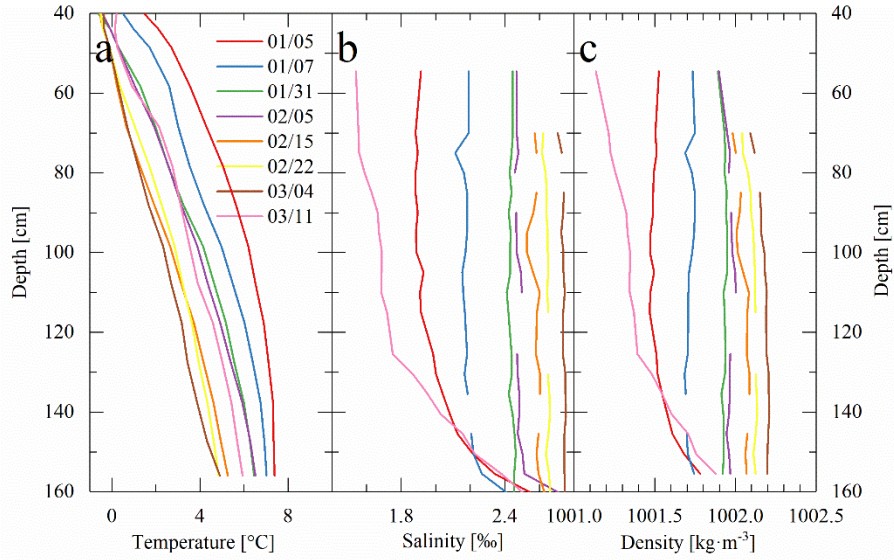

**Figure 6. Observed temperature and salinity profiles and estimated water density (according to Leppäranta (2015)) in winter 2017.**

**3.3 Heat transfer at the ice-water interface**

Heat and mass fluxes at the ice-water interface govern the basal freezing/melting rate of ice cover and the temperature of the top water layer. Our data show that in mid-winter, ice growth slowed down, and then a near-equilibrium period appeared (i.e., the thickness kept roughly constant) prior to the start of the melting period (Fig. 4). At the ice bottom, the latent heat flux $Q_l$ kept positive during continuous ice growth and fluctuated near zero level during the near-equilibrium period. Thereafter the ice began to melt from bottom, and $Q_l$ turned negative (Fig. 7). The conductive heat flux $Q_c$ through the bottom ice layer kept positive, indicating upward heat transport. After the ice had started fast melting, $Q_c$ went down to near zero with the ice cover turned into a (quasi-)isothermal state.

The water-to-ice heat flux $F_w$ showed a similar variation with $Q_c$. Physically, $F_w$ is crucially determined by the inverse thermal gradient of the topmost interface layer. The thinner interface layer with the higher

thermal gradient in winters 2016 (temporal average ± standard deviation: 40.8±11.7 W m$^{-2}$) and 2018 (44.9±9.4 W m$^{-2}$) created higher $F_w$ than those in winters 2017 (21.4±12.3 W m$^{-2}$) and 2019 (30.2±9.0 W m$^{-2}$). Interestingly, the convective mixing process increased $F_w$ by 33% in winter 2016 but decreased $F_w$ by 26% in winter 2018 compared with $Fw$ before the convection occurrence, indicating complicated effects of convection.

During the ice growth, both latent heat due to freezing ($Q_l$) and conductive heat from water to ice ($F_w$) need to be taken out by the ice conduction heat ($Q_c$) (Eq. (4)). $Q_c$ was predominantly determined by the ice thickness and surface heat balance (Leppäranta, 2015), so a higher $F_w$ meant lower $Q_l$ and growth rate of ice. Specifically, $F_w$ took up > 65% of $Q_c$ prior to the equilibrium stage (e.g., winters of 2016 and 2017) and > 90% in the equilibrium stage (e.g., winters of 2018 and 2019), the remaining of $Q_c$ was used to take the latent heat of freezing out to the atmosphere through the ice cover, leading to continuous ice growth.

During initial ice melting, the heat transfer from water to ice ($F_w$) was largely conducted through the ice cover ($Q_c$) (70%–80%) and partly used to melt the basal ice ($Q_l$). But during the following fast melting, $Q_c$ was negligible since the isothermal ice cover depresses or even prevented heat conduction and $F_w$ was almost totally used for basal ice melt.

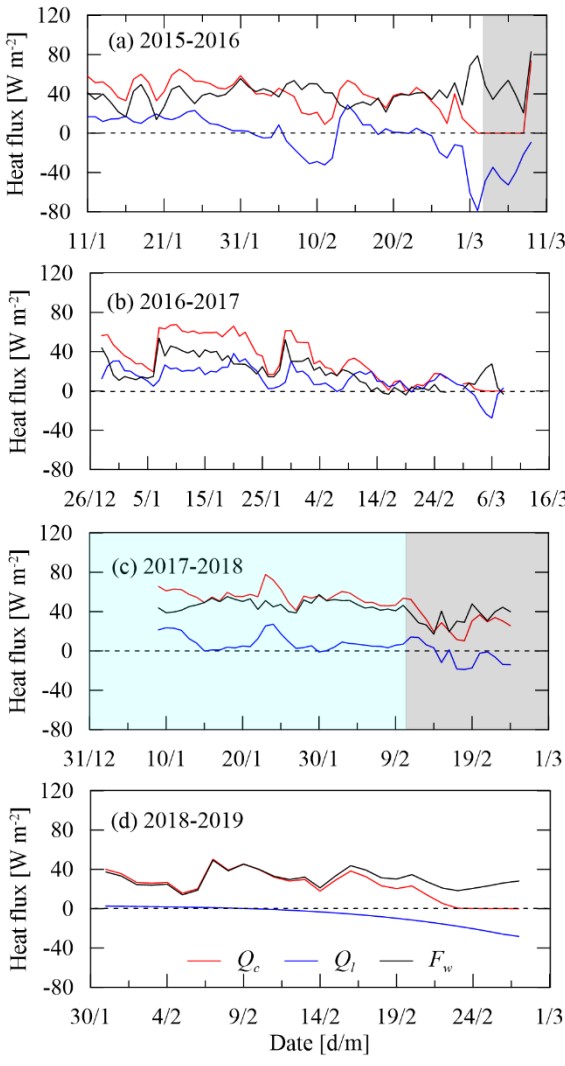

Figure 7. Heat fluxes at the ice-water interface ($Q_c$: conductive heat flux in the bottom ice; $Q_l$: latent heat flux due to basal ice freezing/melting; $F_w$: water-to-ice heat flux). The light gray and blue

**zones denote periods of convective mixing and stratification with the local "warm" layer (Fig. 5), respectively.**

### 3.4 Energetics of the water column

The temperature regime of under-ice water is governed by the heat budget. Fig. 8 shows all the heat fluxes involved and the balance residual. In mid-winter, the solar flux $R_w$ was 25–50 W m$^{-2}$ under bare ice cover and dropped to 1.5–13 W m$^{-2}$ under ice with a snow cover of varying thickness (1.5–8 cm) and age. Only 3%–14% (1–5 W m$^{-2}$) of $R_w$ (i.e., $R_{sed}$) reached the sediment surface (Fig. 4), which in turn released heat ($F_{sed}$) to the overlying water in mid-winter (1–3 W m$^{-2}$). The heat flux from water to ice, $F_w$, also showed interannual and seasonal variations (10–60 W m$^{-2}$) and was generally smaller under snow-covered ice than that under bare ice, likely indicating the effect of transmitted sunlight. The heat content change ($F_{Tw}$) of water, as a resultant heat change from heat sources and sinks, was typically small ($-5 - +4$ W m$^{-2}$) during freezing but grew up to 4–15 W m$^{-2}$ during the initial melt.

Evidently, the transmitted solar radiation ($R_w$) and water-to-ice heat transfer ($F_w$) dominated the heat balance of the under-ice water. Combining the 4-winter observations, $R_w$ was the largest heat source (34.8±18 W m$^{-2}$) and accounted for (92±9)% of the total source ($R_w+F_{sed}$) to the under-ice water, while $F_w$ was the largest heat sink (34.3±15 W m$^{-2}$) and accounted for (96±38)% of the total sink ($F_w + R_{sed}$). The term ($F_{sed} - R_{sed}$) was only –0.8±2.7 W m$^{-2}$ and $F_{Tw}$ was 0.7±8.7 W m$^{-2}$, both of which can be neglected compared to others. Therefore, the transmitted solar radiation was almost totally (97%) returned to the ice base by means of water-to-ice heat conduction.

Inter-annual comparisons indicated that winter 2017 with only a prevailing inverse temperature structure and a decreasing bulk temperature was different from that in other winters (Figs. 4 and 5). Heat flows and budget can provide basic insight into the differences. During the freezing period of winter 2017, the bulk water temperature kept decreasing because the net heat gain of water was negative (i.e. $R_w+F_{sed}-F_w< 0$); Continuous heat loss of water to the ice bottom also created inverse thermal gradient and decrease in water temperature prevented the occurrence of mixing. However, in other winters (especially 2016 and 2018), the net heat gain of water was positive, so the water temperature had an increasing trend, which increases the potential for mixing occurrence. Compared with other winters, snow bands and spots that prevailed on top of a thicker ice cover in winter 2017 caused lower penetrated solar radiation, largely contributing to the general cooling of the water column.

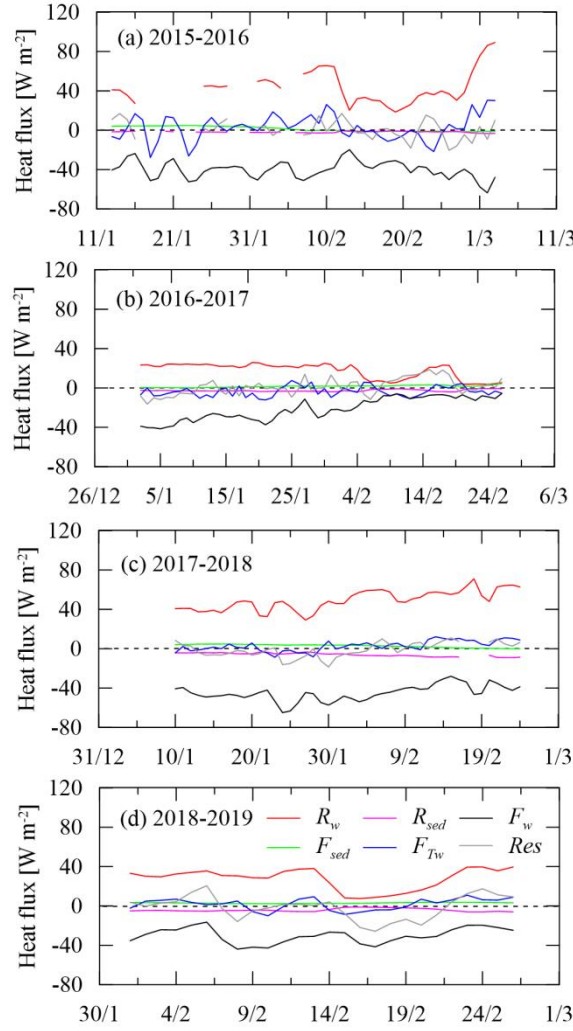

**Figure 8. Heat budget of the under-ice water ($R_w$: transmitted solar radiation; $R_{sed}$: absorbed solar radiation by sediment; $F_w$: water-to-ice heat flux; $F_{sed}$: heat released from sediment; $F_{Tw}$: sensible heat caused by water heat content change; $Res$: residual of heat balancing, which is supposed to be zero when all heat fluxes balance ideally)**

## 4 Discussion

### 4.1 Comparisons with (sub)Arctic and temperate climate lakes

Prior to the ice-on date, in freshwater lakes fall mixing due to thermally free convection (at 3.98°C) and continuous wind stirring against weak salinity gradients usually create large/full-depth vertical isothermal structure with temperature quite close to the freezing point (stage **I** in Fig. 9).

After the freeze-up or ice-on, the under-ice stratification evolves as a joint result of snow and ice condition, solar radiation penetration into water, heat flux from the bottom sediment, and horizontal advection and diffusion. In Arctic, boreal, and northern temperate regions, such as Fennoscandia, north America, and central Europe, winter precipitation leads to thick snow cover on lake ice, and only little sunlight can penetrate through the snow and ice cover and, hence, can be neglected in the water column. The water column receives heat from the bottom sediment and releases heat to the ice cover on top. These heat fluxes are small (0–5 W m$^{-2}$), and therefore the lake water stays close to the freezing point in the top layer and has a very weak inverse structure (curve **I**) through the entire growth period of 3–5 months.

After the melting onset, warm air and strengthened solar radiation lead to snow melting, and more solar radiation goes through the transparent ice and heats up the underlying water, creating a deepening convective mixing (stage **II**) before reaching the temperature of maximum density ($T_m$) (stage **III**). Usually, the ice cover breaks up before the thermal state of stage **III** forms in most deep boreal and Arctic lakes (Yang et al., 2020).

In mid-latitude cold and arid regions, intensive solar radiation and thin snow cover allow more solar energy transmittance to the water column just following the freeze-up. In the Qinghai-Tibet Plateau (QTP), the water column can keep a stable state of stage **I** or start slow warming (i.e., period of stage **II**) just following the freeze-up in deep lakes, and then go to stage **III**, creating mid-winter overturn (Fig. 9b). Afterwards, strong solar radiation due to thin ice warms continuously the top water layer (stage **IV**), which exists for 4-6 weeks before breakup (Kirillin et al., 2021; Lazhu et al., 2021). However, in shallow ponds, stage **II** (i.e., transition from stage **I** to **III**) is very short (one week), and the water column roughly stays at stage **III** almost over the entire freezing period. The following warm layer (**IV**) can deepen to near the lake bottom before ice-off (Fig. 9c) (Huang et al., 2019b).

Lake Ulansuhai is very shallow and weakly saline, and although the solar radiation is strong, thermal stratification dynamics is determined by the synchronous profile evolution of temperature and salinity. Although our observations covered only the mid-winter, the thermal profile of type **I** is expected at the pre-winter and ice-on due to joint effects of wind-stirring and salinity gradient. But stage **I** should be very short, and the bulk temperature increases rapidly and transition to stage **II** takes place due to the solar radiation transmittance and shallow lake depth. However, the occurrence of convective mixing (we used stage **IIIb** here for brackish water) is conditional and mainly dependent upon the salinity evolution due to the freezing-exclusion effect. Stage **IV** is also expected since meltwater dilution in the top layer can suppress the convection. Note that, the forming regime of stage **IIIb** is different in this brackish lake compared with stage **III** in freshwater lakes, which are predominantly driven by temperature approaching $T_m$ with solar heating. In brackish lakes, convective mixing may be stopped by a dicothermal layer in the middle (Fig. 5c) and full convection is possible only when the bottom water is warm enough to conquer the salt stratification.

Salinity structure plays a more important role in lake stratification and convective mixing than the temperature in brackish/saline and even freshwater lakes with salinity below 0.5 ppt (Kirillin and Terzhevik, 2011). The present results indicated that the salt exclusion during freezing changes both the total salt content and salinity structure. For instance, for a lake with a mean depth of 1.0 m, if the salinity segregation coefficient is assumed 0.15 (Pieters and Lawrence, 2009; Bluteau et al, 2017), formation of 0.5 m ice cover can cause an increment of 70% to the water salinity. In Lake Ulansuhai, the salinity increases downward at the ice-on with a large salinity gradient. Afterwards, as the ice grows, salt exclusion gradually decreases the salinity gradient, making the water more prone to mix convectively.

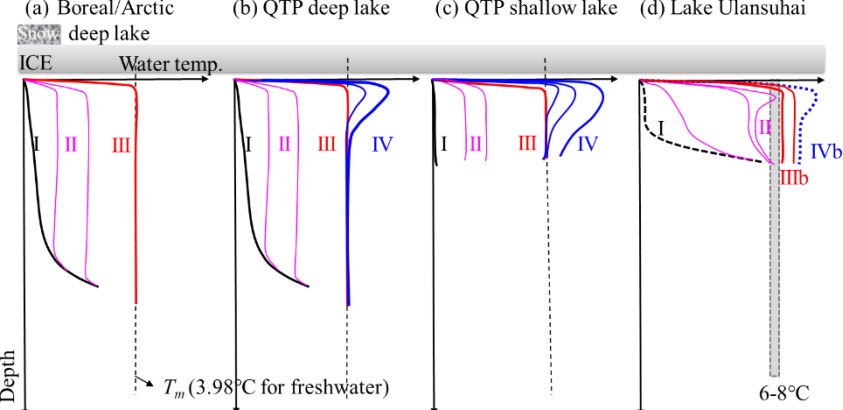

**Figure 9. Typical thermal stratification types in ice-covered lakes: (a) deep lakes in Arctic (Jakkila et al., 2009), (b) deep lakes in QTP (Kirillin et al., 2021; Lazhu et al., 2021), (c) a shallow pond in QTP (Huang et al., 2019b), and (d) Lake Ulansuhai. The definitions of Roman numbers are presented in the text.**

**4.2 What leads to high water-to-ice heat flux?**

The water-to-ice heat flux $F_w$ plays a predominant role in the basal growth and melting of lake ice cover but is quite challenging to observe instrumentally. Eqs. (1) and (4) provide two ways to estimate $F_w$ indirectly if the ice thickness, temperature profile of the ice-water-sediment column, and solar irradiance are observed (actually these variables were often observed in lake thermal regime and ice programs).

By definition, $F_w$ is the conductive heat flux across the very thin diffusive water layer just beneath the ice. The temperature gradient and thickness of this thin layer are influenced to a varied extent by thermal stratification, convective mixing (Figs. 5 and 6), advection (Rizk et al., 2014; Kirillin et al., 2015), and seiche oscillation (Kirillin et al., 2018). All these thermal and hydraulic processes lead to non-stationary and spatiotemporally varying $F_w$ (Winters et al., 2019).

In freshwater lakes, under-ice convective mixing is observed to increase heat transport to the ice bottom by increasing the thermal gradient of the interfacial layer above the convective layer (Mironov et al., 2002; Kirillin et al., 2018). However, in weakly saline Lake Ulansuhai, under-ice convective mixing does not necessarily take place every winter and its impact on heat transport to the ice bottom differs annually. In the winter of 2016, the convective mixing developed even across the entire water column, and then encroached the overlying interfacial diffusive layer and increased the bottom temperature of this layer (Fig 5a), resulting in an increase in the thermal gradient of this layer and thus enhancing the heat diffusion (i.e., increasing $F_w$). However, in the winter of 2018, the convection took place only in the lower half of the water column, slightly decreased the thermal gradient of the overlying diffusive layer, and eliminated the dicothermal layer that maintained relatively high $F_w$ prior to the convection onset (Fig 5c), leading to a decrease in $F_w$. In the future, detailed synchronous datasets on synchronous temperature and salinity profiles are needed to understand the accurate regime of convection in this type of lakes.

Although we did not acquire concurrent salinity profiles to the water temperature, sampling results in the winter of 2017 inevitably indicate the development of double diffusion as the temperature destabilizes while the salinity stabilizes the stratification (Schmitt, 1994; Schimid et al., 2010). The effective heat diffusivity of the bulk water column estimated from $F_w$ derived by Eq. (4) was 5–16 (mean of approximately 10) times larger than the molecular diffusivity, indicating the significantly enhanced

diffusivity of heat due to the double diffusion.
In boreal and Arctic lakes, weak solar radiation, short insolation duration, and most importantly thick
snow cover limit solar heat input to the under-ice water column, just water and sediment heat release
(both very small) can cause only low seasonal values, $F_w < 15$ W m$^{-2}$ (Malm et al., 1997; Jakkila et al.,
2009). However, in arid or mid-latitude lakes with thin snow and/or more intensive solar insolation, $F_w$
can be high, 10–50 W m$^{-2}$ in Lake Baikal (Aslamov et al., 2017) and 20–100 W m$^{-2}$ in QTP lakes with
distinct seasonal variation (Huang et al., 2019a,b; Kirillin et al., 2021). The estimated $F_w$ in Lake
Ulansuhai is comparable to Lake Baikal and QTP lakes, indicative of the vital contribution of solar
radiation and the absence of snow cover.
Higher $F_w$ does not necessarily mean growth suspend, shorter freezing duration, or thinner lake ice cover.
In Lake Ulansuhai, in ice growth, the conductive heat in the ice cover ($Q_c$) is much higher, which means
that the $F_w$ can be totally released through the ice cover and the freezing latent heat ($Q_l$) can also be taken
out since $F_w + Q_l = Q_c$. This ensures the continuous growth of ice. In ice melting, the $Q_c$ is usually
ignorable, the under-ice water supplies heat ($F_w$) to maintain basal ice melting ($Q_l$). Higher $F_w$ means a
greater melting rate.
From a perspective of heat balance in water (Eq. 1),
$$F_w = R_w - R_{sed} + F_{sed} + F_h - \rho_w c_w h_w \frac{dT_w}{dt} - \rho_w c_w T_w \frac{dh_w}{dt}, \quad (5)$$
If we define $Q_{rad} = R_w - R_{sed}$ (i.e., solar absorption by the water column), and the heat content change
due to subsurface water seepage is negligible, Eq. (5) is transformed to
$$F_w = Q_{rad} + F_{sed} + F_h - F_{Tw}, \quad (6)$$
which means the solar energy ($Q_{rad}$) and sediment heat ($F_{sed}$) are used to change the bulk water
temperature ($F_{Tw}$) and its structure. In turn, the water body loses heat to the ice by adjusting its bulk
temperature and structure. $R_{sed}$ is usually very small, so,
$$Q_{rad} \approx R_w. \quad (7)$$
And Eq. (6) can be transformed to a simple linear formula to present the contribution of $R_w$,
$$F_w = aR_w + b, \quad (8)$$
where slope $a$ reflects the contribution of the penetrated solar radiation while intercept $b$ reflects the
integrated contributions of other heats. Fig. 8 argued that both $F_{sed}$ and $F_{Tw}$ are very small and roughly
constant and $R_w$ and $F_w$ are the overwhelming dominant heat source and sink, respectively. In
consequence, if we fit the $F_w \sim R_w$ data using $a=1$, the regressed mean contribution of heat fluxes except
is –3.1 W m$^{-2}$ (red line in Fig. 10), very close to the estimate of –1.5 W m$^{-2}$ in Section 3.4. If we ignore
the minor intercept, the line with $a=0.93$ explains approximately the same amount of variance in the
observations (blue line in Fig. 10), consistent with the observed ratio of $F_w$ to $R_w$ (0.97).
But we have to note that values of both coefficients should be lake specific. Lake depth and salinity
modify the changes in convective mixing depth, bulk water temperature, and temperature structure
caused by solar irradiance (Lazhu et al., 2021), and thus alter the relative contributions of solar radiation
to water heat content and to heat transfer from water to ice. For instance, in a deep lake with a mean
depth of 20 m in Finland, 1/3 of the transmitted solar radiation returned to ice (Leppäranta et al., 2019).

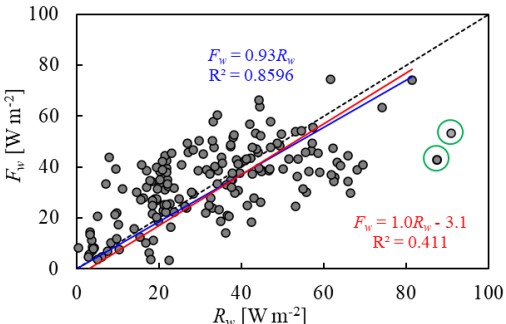

**Figure 10. Linear fitting of daily water-to-ice heat flux $F_w$ as a function of penetrated solar radiation $R_w$. The black dashed line denotes $F_w = R_w$. Two models were used to fit the data with two dots in green circles being removed.**

### 5 Conclusions

We present the ice-covered lake thermodynamics in a mid-latitude, cold and dry region, where the climatic and hydrological environment is in distinct contrast to the Arctic, boreal, and other northern temperate regions. The ice cover is always bare or covered by only occasional thin snow patches lasting for 1–2 weeks due to the arid climate and wind blowing. The clear congelation ice cover allows 1/5–1/3 of incident solar radiation to penetrate into the water column in mid-winter, providing a background for the energetics of under-ice water. The transmitted radiation and heat transfer across the ice-water interface dominate the heat budget of the water column and are highly correlated. High water-to-ice heat flux $F_w$ (daily averages of 20–45 W m$^{-2}$) was observed compared to that in (sub)Arctic and boreal lakes and takes up >90% of the solar radiation input to the under-ice water (20–50 W m$^{-2}$). Snow accumulations can decrease $F_w$ due to its large albedo and light attenuation. Despite of high $F_w$, higher heat conduction within the ice cover (30–55 W m$^{-2}$) existed during the freezing period because of the persistent snow-free ice surface and created continuous basal growth of ice. In particular, the high correlation between $F_w$ and penetrating solar radiation indicates the temporal variation of $F_w$, which is important for updating $F_w$ parameterization in lake ice modelling.

Both bulk water temperature and its structure show diurnal, synoptical, and seasonal variations due to their quick responses to transmitted radiation and snow events because of the small lake depth. Double diffusion should surely prevail in wintertime in this shallow saline lake and strengthens the heat transport to the ice bottom because there is always cooler and fresher water overlying warmer and saltier water. Under-ice convective mixing and/or dicothermal water layer take place in some winters depending on the dynamic interaction between radiation (temperature) and salinity stratifications, which is mediated by the salt exclusion during freezing. However, details in double diffusion, convective mixing, and the effect of salt exclusion (or cryoconcentration) on water stratification in shallow ice-covered saline lakes need to be investigated in the future using high-frequency and high-resolution measurements.

*Data availability.* The main datasets on lake ice/snow thickness, temperatures, and transmitted solar radiation used in this paper are available at https://zenodo.org/record/4291840 (doi: 10.5281/zenodo.4291840).

*Author contributions.* WH, ML, and ZLi conceived the study. WZ, HY, and ZLin conducted the field observations. WZ, CZ, RL, and ZLi analysed data on meteorology and ice/snow conditions. WH and ML developed and ran the model. WZ, RL, and WH calculated the heat budgets for the water column. WH

and WZ wrote the paper with contributions from all the co-authors.

*Competing interests.* The authors declare no competing interests.

*Acknowledgements.* This study was funded by the National Key Research and Development Program of
China (2019YFE0197600), National Natural Science Foundation of China (51979024), the Open Fund
of State Key Laboratory of Frozen Soil Engineering (SKLFSE201813), the Program of Introducing
Talents of Discipline to Universities (B08039), the Fundamental Research Funds for the Central
Universities (CHD) (300102291507), and Academy of Finland (333889). We are grateful to the
technicians of the National Ecologic Station in Lake Ulansuhai and the rest of our field team for their
invaluable help in field campaigns.

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
