# Peer review of "Sunlight penetration dominates the thermal regime and"

_The Cryosphere, 2021_

## Referee Comment (RC2)

The study is dedicated to the heat budget of ice-covered waters in Central Asia. This is a weakly investigated topic important for understanding the seasonal ice balance in large arid endorheic regions of strongly continental climates.The authors used four years long observations of temperature and radiation in a large shallow lake of Inner Mongolia. High temporal and vertical resolution of observations allowed estimation of the boundary fluxes in the ice-water-sediment system and their relationship to solar irradiance, and ice and snow thicknesses. Data from the ice-covered seasons from 2015 to 2019 provided estimates of inter-annual variability in the winter heat budget. The methods are generally correct and adequate to the posed research questions. The results are of interest for the ice research community and are suitable for publication in "The Cryosphere". The manuscript is well-organized. The presentation can be however improved by language and style editing.

I have some remarks and questions on the analysis of the results. In particular, the concluding part of Discussion, including Eq. 6 and Figure 9, is rather confusing. Why apply a least-square linear model to approximate the water-ice heat flux $F_w$ as a function of solar radiation $Q_{rad}$? It directly follows from your data that $F_w \approx Q_{rad}$ (see the last sentence before Eq. 6). Hence, the coefficients $a$ and $b$ in your linear model have no physical meaning, unless you propose their interpretation. Moreover, looking at Fig. 9, one could suggest that a straight line $F_w = Q_{rad}$ would explain approximately the same amount of variance in the observations (see the blue line in the drawing below), especially if the outliers at very high under-ice radiation levels (blue circles in the drawing) are removed. Herewith, apart from being unjustified physically, the coefficients $a$ and $b$ introduce only additional uncertainty without any additional predictive power. This part of the analysis requires essential revising.

[Figure]

Below are remaining comments and questions arranged along the text flow.

- Line 104: Figure 1a needs some edits and(or) explanations. What do the colored areas mean? They are subscribed in Chinese only. If it is a classification of climatic zones, where it comes from? A reference to the source is needed.

- Lines 115-117: I have not found any information on water depths where

the irradiance sensors were installed in the water column.

- Line 137: Eq. 1 is valid if $T_w$ is the water temperature averaged across the water column. It should be explicitly stated in the text.

- Line 145: How the extinction coefficient was measured?

- Lines 152-158: Can you provide details on the "optimal control model"? How deep the temperature loggers were buried in the sediments? How the thermal conductivity of the sediment was estimated?

- Lines 163-165: Replace "first" with "second" and vice versa.

- Lines 176-177: Why these certain thresholds were chosen for the irradiance? Can you compare them to typical seasonal radiation values under ice?

- Line 192: Did absolute humidity change in the diurnal cycle, or was it just an effect of the air temperature variations?

- Line 218: Replace "persist" with "persistent"

- Lines 223-225: It would be more consistent to describe the phenomenon as a local temperature *minimum* created by vertical salinity gradient preventing downward heat transport from the upper waters. Cf. Mironov et al. [2002, Section 6 "Effect of salinity"].

- Lines 242-243: The temperature-salinity distribution described here inevitably suggest development of double-diffusive convection [Schmitt, 1994]. While the existing data do probably not allow direct estimations of double diffusion, its potential role in the vertical heat transport is worth mentioning here or in Discussion.

- Lines 269-271: How the relative contribution of convection to $F_w$ was estimated?

- Lines 359-361: Eq. 3 requires temperature profiles within the ice cover and knowledge of the heat conduction coefficient. Neither of them are "routinely observed".

**References**

D. Mironov, A. Terzhevik, G. Kirillin, T. Jonas, J. Malm, and D. Farmer. Radiatively-driven convection in ice-covered lakes: Observations, scaling and mixed-layer model. *Journal of Geophysical Research*, 107(C4):7–1–7–16, 2002. doi: 10.1029/2001JC000892.

R. W. Schmitt. Double diffusion in oceanography. *Annual Review of Fluid Mechanics*, 26(1):255–285, 1994.

---

## Author Comment (AC1)

**Reply to RC1**

Note: the **comments** and **authors' replies** are in font color of **black** and **blue**, respectively.

The manuscript (TC-2021-349): "Radiative penetration dominates the thermal regime and energetics of a shallow ice-covered lake in an arid climate" by Huang et al., presented temperature and radiative flux observations in a shallow lake during the ice season. The lake was quite big and located in the Mongolian Plateau northwest China where the local weather was dominated by an arid climate. The observations have been made for 4 winter seasons. For each season, the observation covered between 1-2 months started from January to February or March. The heat fluxes across the ice-water interface and distributed within the lake water below the ice bottom were investigated. The paper concluded that the penetrated solar radiative flux contributed significantly to warming the lake water and creating large lake water heat flux below the ice bottom.

The shallow lake energy budget during winter is important. it plays an important role in lake water-bottom sediment interaction and eventually affects the lake or the surrounding watershed environment during the ice-free season. I found this research topic fits very well the scope of TC journal. The observations technique and data analyses presented in this manuscript are in general adequate. The method is sound, and the results are convincing. However, I found issues with respect to lack of clarity and weak causal presentations in various places in the manuscript. I think extra work is needed to improve the overall quality of this manuscript before it can be considered as TC publication. Please see my comments point by point below that I hope can be useful for authors to improve the manuscript.

Reply: Thanks for your detailed consideration and constructive comments. We took your comments in deep consideration and revised our manuscript. Your advices improved the total quality of the manuscript.

1: The title of the manuscript makes me feel this study is rather qualitative. You might consider reformulating the text and making it look more quantitative, e.g. The impact of solar radiative flux on the thermal regimes and energetics of a shallow lake in an arid climate region during winter or something like that.

Reply: We modified it to *Sunlight penetration dominates the thermal regime and energetics of a shallow ice-covered lake in arid climate.* The verb *dominate* may sound qualitative somehow, but this title we guess presents the core finding of the present work since the stronger incident solar radiation and the absence of snow cover in arid region lake allow for more sunlight transmission that provides background energy for heat flows in this ice-covered lake, different with arctic or boreal lakes where sediment heat release may play a key role.

2: In the abstract, "The Central Asia" sounds like a very large area, I am sure there are papers focused on winter lake studies. It might be better to be more specific to point out your research domain, i.e., Mongolian Plateau

Reply: Your suggestion is good; Mongolian Plateau is more specific.

3: L20, "Results reveal that persistent bare ice permits 20%–35% of incident solar radiation…" is this range independent of bare ice thickness? please specify.

Reply: Based on the existing ice thickness (during the observation period), it allowed 20-35% of the solar incident radiation to penetrate the ice. So, we can see the value of $F_w$ is changing. Theoretically, transmitted radiation is related to ice thickness. In general, as the ice thickness grows continually, the transmitted radiation decreases. But during our observed periods, the ice thickness did not change much (35-50 cm), the weakening of the radiation by the ice is relatively stable. And we added this thickness range to this statement.

4: L22/23, "high water-to-ice heat flux (annually mean 20–45 W m-2) in mid-winter" What do you mean "annually mean" (annual mean)? How did you define the annual cycle? Your observations covered only partial winter, so "annual mean" is a bit misleading. Please revise accordingly.

Reply: Thanks for your reminding. As you said, annual mean is not a correct definition here. The value is actually the average over the observational period (i.e. mid-winter).

5: After I read the entire manuscript, I felt you might consider adding more findings you have discovered in this study, for example, the heat flux within ice floe could be summarized in the abstract along with the water-to-ice heat flux.

Reply: Your advice reminds us of one of the new findings in our work. Besides the high water-to-ice heat flux, the heat conduction within the ice interior is also remarkably high compared to previous findings in arctic and boreal lakes. More importantly, the high in-ice heat conduction makes the ice keep growing despite the high water-to-ice heat flux. We added this information in the revised abstract.

6: Introduction: Please consider reformulating the last paragraph. I think the first half paragraph may suit better in the following Chapter. The second half of this paragraph looked like the objectives of this study. You may start with a discussion on what subjects or knowledge gaps were missing for the lake energy balance study. Then point out the objectives of this work.

Reply: Thanks for your suggestion. The knowledge gaps were mainly the lack of lake ice and ice-covered lake thermodynamics in cold and arid central Asia and were discussed/concluded in the preceding paragraphs. To make it more focused, we rephrased this paragraph to "*To fill the knowledge gaps in winter thermodynamics of lakes in cold and arid Asia and their background energy flows, we performed a four-winter observation program of snow/ice processes, solar radiation transfer, and temperature profiles of the air-ice-water-sediment column in a typical large shallow*

*lake that is seasonally ice-covered for 4–5 months, located in the southern border of the Mongolia Plateau. Below, observations and models are combined 1) to reveal the seasonal and diurnal dynamics of the temperature stratification under ice in mid-latitude arid climate, and 2) to quantify and balance the involved heat fluxes that determine the thermal state of the lake.*"

7: Chapter 2 is named "Method". I would suggest you reconsider the title of this chapter, e.g. "Data and method" perhaps better.
Reply: Revised accordingly.

8: Section 2.1 "study site" is kind of ok, but could you give a bit more information on the study site that is connected with the water-energy budget. I mean to provide some more information on the importance of lake water energy study for this particular region.
Reply: Thanks for your advice. We added general information on the importance of lake water energy study for this region, reading "*Due to its shallow depth and eutrophication, the lake ecology under the ice cover is very active with high rates of primary production and respiration. This is believed to be highly related to the under-ice solar irradiance and temperature and the key role of ice and snow processes (Song et al., 2019; Huang et al., 2021). Our previous observations revealed the mass and heat balance of the lake ice cover and the impacts of warm water under the ice cover (Lu et al., 2020), but further investigations were performed and combined here to look into the thermal stratification regimes.*"

9: I see Figure 1 is fine in terms of illustration. However, the figure caption is not very informative and need to be reformulated. For example, there is no information in figure 1c.
Reply: Thanks for your reminding. We added general description of the field instruments deployed in figure 1c.

10: Section 2.2 is ok. But it would be more interesting to describe field observations to some extend with text and even photos rather than list a bunch of numbers in a Table.
Reply: Thanks for your good suggestion. In table 1, we changed the numbers of deployed sensors to the sensor depths (i.e. distances below the ice surface). And we added a new figure (Figure 2) to present a picture of *in situ* setups of sensors in the winter of 2019.

11: How do I understand the "total number of measuring depths showed in the bracket" in the third line of Table1 (Air-ice-water-sediment temperature)? What electric conductivity means here? How do you use it in this study?
Reply: The "total number of measuring depths" and numbers in blankets may cause misleading here, so we changed them to the real observation depths (i.e. the distances below the ice surface) of thermistors, spectroradiometers, and conductivity sensors.

The electric conductivity (unit: mS m$^{-1}$) is roughly also a measure of water salinity and is easier to observe in situ continuously than salinity. The electric conductivity can be used to calculate the salinity based on some empirical formulas and to estimate the water density structure here (e.g. Figure 5) by combining with water temperature.

12: Section 2.3 is better entitled as "method".
Reply: Thanks for your advice. We changed the title to "Heat flux calculation and balance", which we think is more concrete.

13: Figure 2 is nice, but I have seen it in a published paper by (Huang et al., 2019, https://doi.org/10.1016/j.jhydrol.2019.124122). Although Huang is also the first author of this manuscript, it might be better to cite the original source of this figure or make the necessary edition of Figure 2 accordingly.
Reply: Thanks for your reminding. We added the source to the figure caption and actually this figure was modified from the source figure.

14: L176, We classified errors into four ranges: please give a citation or add some arguments on such classifications. In addition, please check the notations of table 2.
Reply: These ranges were set following Lei et al (2014) on multi-year Arctic sea ice. Its definition seems somehow arbitrary but does provide a measure/scale for error analysis.
Comments on this part from you and Reviewer #2 make us realize that grading the error into four ranges is arbitrary and not necessary in the context, and just giving the error values is enough to evaluate the uncertainties in heat flux calculations. So, in the revised version, we removed the four categories and just kept the error values in this section.

15: Figure 3 is nice and informative. Please consider using international standard [dd/mm/yyyy] as the x-labels. Please check the figure legends: should it be the "transmitted" or "reflected" as in the figure caption.
Reply: We used international standard date format. Additionally, it is "transmitted".

16: In section 3.2, it would be better to discuss what do you know about the salinity stratification during your observational periods in order to support your statement on unconventional thermal stratification. How did you measure lake water salinity?
Reply: Thanks. In winter 2017, we used 3 electric conductivity (EC) sensors to measure the water EC (can be converted to salinity following empirical formulas) and also sampled ice and water to test their salinity 8 times. The sampling results were shown in Fig. 6 and used to calculate the water density. We added our observational methods on water EC to section 2.2 (reading "*In winter of 2017, the under-ice water electric conductivity (EC) was measured using 3 online conductivity loggers (HOBO U24, Onset, USA) at depths of 60 cm, 100 cm, and 150 cm from Jan 20 to Mar 12. Concurrently, ice and water samples with 5 cm spacing were collected 8 times this winter to measure their EC and salinity using a portable YSI salinometer*") and results of water salinity to the first paragraph of section 3.2.

17: L236-246, Maybe you can argue that the water salinity seems to dominate the stratification of lake water below the ice and penetrating solar radiation below the ice layer as well as the heat flux from sediment determine the lake temperature profile? Why does winter 2017 differ from other seasons?

Reply: By this paragraph and Fig. 5, we were not intended to argue the difference of winter 2017, we just took this winter as an example to show how the salinity and temperature evolved and how they jointly determined the water stratification or mixing because we sampled water to measure its salinity only in this winter, which allowed us to estimate the water density profiles. We added this information to this part.

If we look into the difference of winter 2017, during the freezing period in this winter, (1) the ice grew faster because the residual of (Qc-Fw) gave higher latent heat (Ql) compared to other winters; (2) the bulk water temperature kept decreasing because the net heat gain was negative (i.e. Rw+Fsed-Fw); (3) the temperature structure was somehow different with prevailing inverse temperature profiles and without apparent mixing, this is likely because the continuous heat loss of water to the ice bottom (negative heat gain as abovementioned) creates inverse thermal gradient and decrease in water temperature, which prevents the occurrence of mixing. However, in other winters (especially 2016 and 2018), the net heat gain of water is overall positive, so the water temperature had an increasing trend, which increases the potential for mixing occurrence.

We added this information (L345-352).

18: Figure 4 needs makeup. The current illustration is too messy and difficult to see clearly.

Reply:  Thanks for your suggestion. We replotted it to make it more concise and clear, as below,

[Figure]

Figure 4. Daily profile evolution of water column during ice season of winters (a) 2016, (b) 2017, (c)

2018, and (d) 2019. Light blue and brown zones denote ice cover and bottom sediment, respectively. Asterisks (※) denote snowfalls and snow-covered periods. Plus (+) and minus (-) denote growth and melt stage of the ice cover.

19: I think section 3.3 is a very important part of this study. I would like to see more discussions. For example, how did you get those numbers in the second paragraph? What are those numbers after the symbol "±", although I can make a pretty guess of them, you need to tell the readers?

Reply: Sorry, those numbers with the symbol "±" denote "a temporal average ± standard deviation", we added this. We also added more information on the physical relations between these fluxes in this section and Discussion, reading "*Higher $F_w$ does not necessarily mean growth suspend, shorter freezing duration or thinner lake ice cover. In Lake Ulansuhai, in ice growth, the conductive heat in the ice cover ($Q_c$) is much higher, which means that the $F_w$ can be totally released through the ice cover and the freezing latent heat ($Q_l$) can also be taken out since $F_w + Q_l = Q_c$. This ensures the continuous growth of ice*".

20: L257-269, Figure 5 presents the temperature and salinity profiles in winter 2017. Please explain why under-ice mixing took place while the convection didn't. Please provide more details on how does the temperature-salinity interact with each other and whether the interaction could trigger convection or not?

Reply: Thanks for your suggestion. In this manuscript, the convection is generally equivalent to vertical mixing by its physic meaning. It is not a simple regime of how the temperature-salinity interaction drives or triggers the convection or not, especially in saline water. In shallow lakes during ice-covered period, the convection or mixing is usually caused by the density instability, i.e., denser water overlies lighter water layer. According to the state equation of water, temperature and salinity are key factors to determine the water density. The temperature of the maximum density (Tm) of water in lake Ulansuhai is about 3.6-3.8°C due to its small salt content. The water density gets smaller as the water temperature gets farther from Tm or as the salinity gets smaller. But in winter, the water temperature is usually below 8-10°C, very small salinity changes can compensate for the influence of temperature on the water density (Fig. R1, referring to Figure 2.4 in Lepparant, 2015). So, in our cases, the lower warm water layer tends to destabilize (convectively mix) the water column but the salinity profiles tend to stabilize the water column. When the lower water gets warm enough, its impact on water density exceeds the impact of concurrent salinity stratification, and thus the convection can be triggered.

We added this description to section 3.2 and added discussion on this to Discussion (section 4.2).

[Figure]

Fig. R1 (i.e., Figure 2.4 in Lepparant, 2015). Density of water as a function of temperature for different salinities at zero gauge pressure: (a) Pure water density as a function of temperature, (b) Density difference from the density at 0 °C in fresh and brackish water (salinities 0, 1, 5 and 10 ‰).

Reference:

*Leppäranta, M.: Freezing of lakes and the evolution of their ice cover, Springer, Berlin, Heidelberg, 2015.*

21: L269, "Interestingly, the convective mixing process increased Fw by 33% in winter 2016 but decreased Fw by 26% in winter 2018, indicating a complicated effect of convection. Those increasing and decreasing are compared with what? Please write your results in a causal way.

Reply:  The increase and decrease are compared to Fw value before the occurrence of convective mixing.  We used a relatively simple and course estimation, i.e., we compared the Fw values prior to the onset of convection and during the convection. For instance, if Fw increased when convection started, we think the convection accelerates the water-to-ice heat transfer. We added this information here.

22: L269-271, here authors stated that the convection increases the Fw in one winter but decreases it in another. Please elaborate on this finding and give some more discussions. Usually, the convection is believed to increase the Fw?

Reply: Yes, previous reports indicate that the convective layer usually takes place beneath the thin stratified interfacial layer and above the quiescent layer with a downward temperature increase (but lower than 4°C) in deep freshwater lakes, and is usually believed to increase the Fw majorly by increasing the bottom temperature of the interfacial diffusive layer and minorly by encroaching this layer (Figure 7 in

Mironov et al., 2002; Figure 4 in Kirillin et al, 2018). This is the general regime appearing in freshwater lakes, but in saline lakes, it may be a different story.

In saline lakes, like Lake Ulansuhai, minor change in salinity profile plays more important role in determining whether the water is stably stratified or not and more importantly determining the occurring depth and thickness of the convection with joint impact of temperature. In winter of 2016, the convective layer appeared even across the whole water column, encroached upwardly the interfacial layer and increased the bottom temperature of this layer (Fig 5a), resulting in an increase in thermal gradient of the top interfacial layer and enhancing heat transport to the ice bottom. However, in winter of 2018, the convective layer took place in the lower half of the water column, slightly decreased the thermal gradient of the overlying stratified layer and eliminated the dicothermal layer that maintained relatively high $F_w$ prior to the convection onset (Fig 5c), leading to a decrease in $F_w$. We added this discussion to Discussion (L423-435).

In the future, detailed datasets of synchronous temperature and salinity are needed to understand the accurate regime of convection in this type of lakes.

23: L272-280, This paragraph draws my attention a lot. It gives a very interesting result, again, I would like to see more discussion here. For example, a large $F_w$ is often associated with a large $Q_c$, which means a strong heat transmission from lake water to lake ice at ice bottom rather than a net positive/negative deficit of heat flux at the ice-water interface to create freezing or melting. What would happen if there were a snow layer on top of lake ice?

Reply: We rephrased these paragraphs following your suggestion and added discussions on how large $F_w$ and $Q_c$ interacted to determine basal freezing or melting of the ice cover (L450-455).

When the ice surface is covered by snow, $F_w$ and $Q_c$ should decrease due to its insulating effect and the fact that the snow layer prevents solar radiation from entering the under-ice water. But the decrease is very little in this lake because only occasional snowfalls take place in winter and accumulated snow is very thin (usually <6-8 cm), short-lived (a week), and sparsely distributed.

24: L295, "small ($-5$–4 W m$^{-2}$)" Not clear to me.

Reply: It means from $-5$ to $+4$ W m$^{-2}$, but the negative sign is the same as En-dash. In order to avoid misleading, we changed it to ($-5$ – $+4$ W m$^{-2}$).

25: In section 4.1 you have defined 4 stages (I, II, III, IV). In section 2.3 You mentioned Winter phase I, Winter phase II. Are there any linkages between those groups of definition?

Reply: Yes, they are of the same category. In stage I, the water temperature is cold (< 3.98°C) with a weak inverse thermal stratification during freezing; In stage II, a convective mixing layer forms between the cold interfacial layer and the warm quiescent layer due to increased sunlight penetration after melting begins.

We realized that we used both "stage" and "phase", which can lead to misleading. So we changed them to "stage" for consistency.

27: The conclusion section is too short. I would like to see a better synthesis of your results and a clear and concrete conclusion that can be regarded as take-home information to the lake ice modeller or lake environment researchers
Reply: Thanks, we enriched this part accordingly.

28: The language can still be improved.
Reply: We had an English speaker checking the manuscript before resubmission.

**Other main changes:**

[1] In this shallow brackish lake, we found almost all (97%) of the transmitted solar radiation returns back to the ice bottom. In *Discussion*, we added comparison with a 20-m deep freshwater lake (Kilpisjärvi) in northern Finland, where 1/3 of transmitted radiation returns to ice (Leppäranta et al. 2019). We think the lake depth is a key geometric factor influencing the ratio because larger depth means a thicker mixing layer, where more solar energy is needed to deepen and heat the mixing layer, so, the returning proportion of irradiance decreases.

[2] *Section 4.1*: After discussion among all authors, we realized that stage III in this brackish lake is of different regime to that in freshwater lakes in Kirillin scheme because of the salinity profile. We reworked on this and named it stage IIIb for brackish water lakes because the convective mixing in brackish lakers may be stopped by a dicothermal layer in the middle and full convection is possible only when the bottom water is warm enough.

---

## Author Comment (AC2)

Note: the **comments** and **authors' replies** are in font color of **black** and **blue**, respectively.

**Reply to RC2**

The study is dedicated to the heat budget of ice-covered waters in Central Asia. This is a weakly investigated topic important for understanding the seasonal ice balance in large arid endorheic regions of strongly continental climates. The authors used four years long observations of temperature and radiation in a large shallow lake of Inner Mongolia. High temporal and vertical resolution of observations allowed estimation of the boundary fluxes in the ice-water-sediment system and their relationship to solar irradiance, and ice and snow thicknesses. Data from the ice-covered seasons from 2015 to 2019 provided estimates of inter-annual variability in the winter heat budget. The methods are generally correct and adequate to the posed research questions. The results are of interest for the ice research community and are suitable for publication in "The Cryosphere". The manuscript is well-organized. The presentation can be however improved by language and style editing. I have some remarks and questions on the analysis of the results, listed in the attached file.

I have some remarks and questions on the analysis of the results. In particular, the concluding part of Discussion, including Eq. 6 and Figure 9, is rather confusing. Why apply a least-square linear model to approximate the water-ice heat flux Fw as a function of solar radiation Qrad? It directly follows from your data that Fw ≈ Qrad (see the last sentence before Eq. 6). Hence, the coefficients a and b in your linear model have no physical meaning, unless you propose their interpretation. Moreover, looking at Fig. 9, one could suggest that a straight line Fw = Qrad would explain approximately the same amount of variance in the observations (see the blue line in the drawing below), especially if the outliers at very high under-ice radiation levels (blue circles in the drawing) are removed. Herewith, apart from being unjustified physically, the coefficients a and b introduce only additional uncertainty without any additional predictive power. This part of the analysis requires essential revising.

[Figure]

Reply: Thanks a lot for your deep consideration and constructive advice. Using a least-square linear model, we intended to only present the significant correlation

between water-to-ice flux (Fw) and transmitted solar radiation (Qrad), but as you suggest, we missed the physical background of the fitted function.
Your analysis makes our results clearer. We totally rephrased this section based on stronger physical discussion (Lines 460-483).

Below are remaining comments and questions arranged along the text flow.
• Line 104: Figure 1a needs some edits and(or) explanations. What do the colored areas mean? They are subscribed in Chinese only. If it is a classification of climatic zones, where it comes from? A reference to the source is needed.
Reply: Thanks for your good suggestion. The colored areas mean classification of climatic zones. But we replaced Figure 1a using a climate zone classification map of China, which gives a more accurate zonation. And this map is provided by the website of China Meteorological Administration ([www.cma.gov.cn](www.cma.gov.cn)). Key information was added to the map.

• Lines 115-117: I have not found any information on water depths where 1 the irradiance sensors were installed in the water column.
Reply: Actually, we can see from Table 1 that the number of irradiance sensors deployed was different at four winters, so the sensor depths were different. Since the sensor depth gives important information, we added the sensor depths in Table 1.

• Line 137: Eq. 1 is valid if Tw is the water temperature averaged across the water column. It should be explicitly stated in the text.
Reply: Revised accordingly.

• Line 145: How the extinction coefficient was measured?
Reply: Actually, the extinction coefficient is calculated from the under-ice irradiance ($R_d$) at two depths at least, following $R_d(z_2) = R_d(z_1) \cdot \exp(-\kappa(z_2 - z_1))$, where $R_d(z_1)$ and $R_d(z_2)$ were observed irradiance at depth $z_1$ and $z_2$, respectively.

• Lines 152-158: Can you provide details on the "optimal control model"? How deep the temperature loggers were buried in the sediments? How the thermal conductivity of the sediment was estimated?
Reply: Optimal control model is one of the common methods to retrieve thermal diffusivity of medium if temperature profiles (>= 3 depths) within this medium are measured continuously. The description of this method can be found in Shi et al, 2014, where we used this method to determine the thermal diffusivity coefficient of lake ice cover. It is physically based on the classical one-dimensional heat conduction equation and is well applicable to temperature profiles with obvious temporal variation and with obvious temperature difference between depths.
In winter 2018, four thermistors were buried in the sediment (1 cm, 9cm, 17 cm, and 30 cm below the sediment surface). We used vertical temperature profiles to estimate the thermal diffusivity of the top sediment based on Optimal Control Model. Then thermal conductivity can be determined with measured density and specific heat

capacity of sediment. We added general information on this method to the revised manuscript (Lines 171-178), but detailed description needs long text and many equations. I suggest readers to refer to the following reference or some manuals on this mathematic model.

*Shi, L., Li, Z., Niu, F., Huang, W., Lu, P., Feng, E., Han, H.: Thermal diffusivity of thermokarst lake ice in Beiluhe basin of the Qinghai-Tibet Plateau, Ann. Glaciol., 55(66), 153-158, 2014.*

• Lines 163-165: Replace "first" with "second" and vice versa.
Reply: Revised accordingly.

• Lines 176-177: Why these certain thresholds were chosen for the irradiance? Can you compare them to typical seasonal radiation values under ice?
Reply: Thanks for your advice. Comments on this part from you and Reviewer #1 make us realize that grading the error is not necessary for the manuscript and just giving the error values is enough to evaluate the uncertainties in heat flux calculations. So, in the revised version, we removed the four categories and just kept the error values in this section.

• Line 192: Did absolute humidity change in the diurnal cycle, or was it just an effect of the air temperature variations?
Reply: After we estimated the absolute air humidity based on synchronous relative humidity and air temperature using formulas in Huang et al. (2016), we can also clearly see the diurnal cycle in absolute humidity (with peaks occurring on afternoon).
*Huang, W., Li, R., Han, H., Niu, F., Wu, Q., Wang, W.: Ice processes and surface ablation in a shallow thermokarst lake in the central Qinghai-Tibet Plateau, Annals of Glaciology, 57(71): 20-28, 2016.*

• Line 218: Replace "persist" with "persistent"
Reply: Modified accordingly.

• Lines 223-225: It would be more consistent to describe the phenomenon as a local temperature *minimum* created by vertical salinity gradient preventing downward heat transport from the upper waters. Cf. Mironov et al. [2002, Section 6 "Effect of salinity"].
Reply: Thanks for your reminding. Yes, this local temperature peak (actually a warmer layer than both overlying and underlying layer) was called temperature minimum in Mironov et al (2002), but was also called a temperature *dichotomy* or *dicothermal layer* in Kirillin et al 2011 [*Kirillin, G., & Terzhevik, A. (2011). Thermal instability in freshwater lakes under ice: Effect of salt gradients or solar radiation? Cold Regions Science and Technology, 65(2), 184–190*]. And we guess the terminology *temperature minimum* is easy to bring readers a misunderstanding, so we used temperature dichotomy instead and added direct description on its thickness and variability, reading "*This abnormal layer is sometimes called a local temperature minimum (Mironov et al., 2002) or a "temperature dichotomy" (i.e., a dicothermal*

*layer used in oceanography) (e.g., Kirillin et al., 2011, 2021). Water temperature contours (not shown) revealed that both the bulk temperature and thickness of the dicothermal layer show significant diurnal cycles: its temperature and thickness take up and increase following the solar insolation cycle and decrease or even disappear during night. The developing and extending of this layer also increases the thermal gradient of the overlying interface layer.*"

• Lines 242-243: The temperature-salinity distribution described here inevitably suggest development of double-diffusive convection [Schmitt, 1994]. While the existing data do probably not allow direct estimations of double diffusion, its potential role in the vertical heat transport is worth mentioning here or in Discussion.

Reply: Thanks for your advice. Yes, the temperature-salinity profiles directly suggest double-diffusive convection, i.e., the temperature destabilizes while the salt stabilizes the water column. But staircases cannot possibly form due to its small lake depth and large heat gradient.

Generally, this diffusive regime is believed to apparently enhance the heat diffusivity compared to that of salt. But we could not estimate the double diffusion (such as the density ratio) due to lack of concurrent data of temperature and salinity. But we can assess its impact on heat diffusivity using dataset of winter 2017. The water-to-ice heat flux Fw gives the bulk effective heat diffusivity of the water column of 5-15 times (mean 10 times) larger than the molecular diffusivity, indicating the enhanced heat diffusivity due to double diffusion. We added this statement to the Discussion, reading "*Although we did not acquire concurrent salinity profiles to the water temperature, sampling results in winter of 2017 inevitably indicate the development of double diffusive convection as the temperature destabilizes while the salinity stabilizes the stratification (Schmitt, 1994; Schmid et al., 2010). The effective heat diffusivity of the bulk water column estimated from Fw derived by Eq. (4) was 5–16 (mean of approximately 10) times larger than the molecular diffusivity, indicating the significantly enhanced diffusivity of heat due to double diffusion.*"

• Lines 269-271: How the relative contribution of convection to Fw was estimated?
Reply: We used a relatively simple and course estimation, i.e., we compared the Fw values prior to the onset of convection and during the convection. For instance, if Fw increased when convection started, we think the convection accelerates the water-to-ice heat transfer.

• Lines 359-361: Eq. 3 requires temperature profiles within the ice cover and knowledge of the heat conduction coefficient. Neither of them are "routinely observed".
Reply: Thank you for your reminding. The thermal conductivity/diffusivity coefficient of ice is not often observed, but of clean freshwater ice, it falls in a narrow range of 2.1-2.2 W m$^{-1}$ °C$^{-1}$. You don't have to measure it. Observations of temperature profiles within ice cover are often observed in lake thermodynamic and thermal stratification research, but maybe not in other researches. So, we modified this

statement to "*actually these variables were often observed in lake thermal regime and ice programs*".

References

D. Mironov, A. Terzhevik, G. Kirillin, T. Jonas, J. Malm, and D. Farmer. Radiatively-driven convection in ice-covered lakes: Observations, scaling and mixed-layer model. Journal of Geophysical Research, 107(C4):7–1–7–16, 2002. doi: 10.1029/2001JC000892.

R. W. Schmitt. Double diffusion in oceanography. Annual Review of Fluid Mechanics, 26(1):255–285, 1994.

Reply: Thanks for providing related publications.

**Other main changes:**

[1] In this shallow brackish lake, we found almost all (97%) of the transmitted solar radiation returns back to the ice bottom. In *Discussion*, we added comparison with a 20-m deep freshwater lake (Kilpisjärvi) in northern Finland, where 1/3 of transmitted radiation returns to ice (Leppäranta et al. 2019). We think the lake depth is a key geometric factor influencing the ratio because larger depth means a thicker mixing layer, where more solar energy is needed to deepen and heat the mixing layer, so, the returning proportion of irradiance decreases.

[2] *Section 4.1*: After discussion among all authors, we realized that stage III in this brackish lake is of different regime to that in freshwater lakes in Kirillin scheme because of the salinity profile. We reworked on this and named it stage IIIb for brackish water lakes because the convective mixing in brackish lakers may be stopped by a dicothermal layer in the middle and full convection is possible only when the bottom water is warm enough.

---

## Editor Decision (ED1)

The Mongolian Plateau is characterized by cold and arid winters with very little precipitation (snowfall), strong solar insolation, and dry air. But little is known about the thermal regimes of ice and ice-covered lakes and their response to the distinct weather and climate in this region. In a typical large, shallow lake, ice and snow processes(cover) and under-ice thermodynamics were monitored for four winters in 2015–2019. Heat transfer at the ice-water interface and the lake heat budget were investigated. The results revealed that persistent bare ice of 35–50 cm thickness  transmits 20–35% of incident solar radiation  into the  water below. This  is a source  of under-ice energy flows and causes/maintains high water temperature (up to 6–8°C) and high heat flux from water to ice (averages of 20–45 W m$^{-2}$) in mid-winter, as well as  higher heat conduction in the ice interior during freezing. The heat balance shows that the transmitted radiation and the heat flux from water to ice are the dominant and highly correlated heat flows in the lake. Both bulk water temperature and temperature structure are sensitive to solar transmittance and occasional snow events. Under-ice convective mixing does not necessarily occur because of stratification of salinity in the water body. In particular, salt exclusion  during freezing changes both the bulk salinity and the salinity profile, which  plays a major role in the stability and mixing of the water column in this shallow lake.

---

## Author Response (AR2)

Note: the **comments** and **authors' replies** are in font color of **black** and **blue**, respectively. All changes in revised manuscript are highlighted using yellow background.

**RC1**

I see the authors made a major revision of the original manuscript addressed all of my comments and criticism. I am satisfied with this new version of the manuscript. I recommend it to be published. Maybe better for authors to check carefully one more time the language, figure and table captions and notations of equations.

I do not have further comments. Well done, nice work,

Reply: Thanks a lot for your suggestion of acceptance. We have checked carefully the manuscript to erase the mistakes/confusions in language, captions of figures and tables.

**RC2**

Here are some minor suggestions on the revised manuscript.

Line 132: I recommend adding the accuracy and resolution of measurement instruments (HOBO Loggers, YSI salinometer etc.) to the text or to Table 1.

Reply: We added accuracy to the text.

Line 233: it is better to use g/kg instead of PSU for salinity, because PSU definition suggests oceanic salt composition

Reply: Thanks, the unit g/kg is the same as PPT or ‰, so we changed it to ‰.

Line 345-352: What are the external factors leading to the specific heat budget in 2017? From Fig. 8 one can conclude that the under-ice solar radiation was lower than in other years. Was it the main reason for the general cooling of the water column? Was the low light penetration caused by the snow cover or by other reasons? The information is missing in the paragraph.

Reply: Thanks for your reminding. Yes, the general cooling of the water column was caused by the negative heat balance, I mean the penetrated solar radiation is smaller than the heat release by Fw. The low light was caused by the snow cover, as well as the thicker ice cover compared with other winters since the ice also absorbs solar energy. We added this info to the paragraph.

Line 478: make the word "Lake" lowercase

Reply: Corrected accordingly.